# Structural Characterization and Anti-Nonalcoholic Fatty Liver Effect of High-Sulfated *Ulva pertusa* Polysaccharide

**DOI:** 10.3390/ph16010062

**Published:** 2022-12-31

**Authors:** Yuzhou Wan, Lin Liu, Bo Zhang, Shaopeng Wang, Xiaoqian Wang, Kexu Chen, Yuxi Li, Tingting Zhao, Huimin Qi

**Affiliations:** 1College of Pharmacy, Weifang Medical University, No. 7166 Baotong Road, Weifang 261053, China; 2Beijing Key Lab for Immune-Mediated Inflammatory Diseases, Institute of Clinical Medical Sciences, China-Japan Friendship Hospital, Beijing 100029, China; 3Department of Pharmacy, Dezhou People’s Hospital, No. 1166 Dongfanghong West Road, Dezhou 253000, China

**Keywords:** *Ulva pertusa*, polysaccharide, structural characterization, nonalcoholic fatty liver disease

## Abstract

The high-sulfated derivative of Ulva pertusa polysaccharide (HU), with unclear structure, has better anti-hyperlipidmia activity than U pertusa polysaccharide ulvan (U). In this study, we explore the main structure of HU and its therapeutic effect against nonalcoholic fatty liver disease (NAFLD). The main structure of HU was elucidated using FT-IR and NMR (13C, 1H, COSY, HSQC, HMBC). The anti-NAFLD activity of HU was explored using the high-fat diet mouse model to detect indicators of blood lipid and liver function and observe the pathologic changes in epididymal fat and the liver. Results showed that HU had these main structural fragments: →4)-β-D-Glcp(1→4)-α-L-Rhap2,3S(1→; →4)-α-L-Rhap3S(1→4)-β-D-Xylp2,3S(1→; →4)-α-L-Rhap3S(1→4)-β-D-Xylp(1→; →4)-α-L-IdopA3S(1→4)-α-L-Rhap3S(1→; →4)-β-D-GlcpA(1→3)-α-L-Rhap(1→; →4)-α-L-IdopA3S(1→4)-β-D-Glcp3Me(1→; →4)-β-D-Xylp2,3S(1→4)-α-L-IdopA3S(1→; and →4)-β-D-Xylp(1→4)-α-L-IdopA3S(1→. Treatment results indicated that HU markedly decreased levels of TC, LDL-C, TG, and AST. Furthermore, lipid droplets in the liver were reduced, and the abnormal enlargement of epididymal fat cells was suppressed. Thus, HU appears to have a protective effect on the development of NAFLD.

## 1. Introduction

The green seaweed *Ulva pertusa* (Chlorophyta) is a main contributor to microalgal bloom, or green tide, a serious marine environmental problem in coastal regions [1,2]. Improving the application of *U pertusa* can reduce the damage of green tide to the marine ecology. In recent years, various uses of green algae, such as for dietary supplementation, have been investigated. *U pertusa* polysaccharide ulvan (U), a rich biologic component in *U pertusa*, has been found to have antioxidant [3], immune-modulatory [4], anti-hyperlipidemia [5], antiviral [6], and antiradiation [7] activities.

With the prevalence of obesity and metabolic disorder-related diseases, nonalcoholic fatty liver disease (NAFLD), which is characterized by excessive deposition of fat in hepatocytes, has become the most common chronic liver disease with an estimated worldwide prevalence of 25.2% [8]. As the growing population of NAFLD patients ages, this will likely result in higher rates of decompensated cirrhosis, hepatocellular carcinoma, and liver-related deaths [9]. Currently, the first-line treatment for NAFLD is lifestyle modifications, such as through diet and exercise for weight loss. Approved therapeutic drugs for NAFLD are still lacking [10]. However, several pharmaceuticals are under investigation and show promise, such as nuclear receptor-targeting drugs, metabolic hormone agents, antifibrotic drugs, and combination therapies [11].

In recent years, marine algae polysaccharides with high sulfate content have attracted attention in the treatment of NAFLD [12]. For example, Zhang et al. [13] found that liver steatosis and hepatocellular ballooning induced by NAFLD were significantly attenuated by a polysaccharide derived from *Laminaria japonica*. Investigation by Ren et al. [14] revealed that sulfated polysaccharide from *Enteromorpha prolifera* decreased serum glyceral trimyristate level by increasing hydrogen sulfide production, suggesting that it attenuated NAFLD in rats fed a high-fat diet (HFD).

Molecular modification, such as sulfation [15] and molecular weight alteration [16], significantly impact the biologic activities of polysaccharides. Our previous investigation found anti-hyperlipidemic activity of ulvan was improved after increasing the content of sulfate [17]. Thus, we hypothesize that this uniqueness of high-sulfated ulvan (HU) may have therapeutic activity against NAFLD.

In this study, the main sugar residues of HU were characterized by average molecular weight, monosaccharide composition, Fourier transform infrared (FT-IR) spectroscopy, and nuclear magnetic resonance (NMR) analyses to identify for the first time the specific linked forms of sugar residues. In addition, we used the mouse model of NAFLD to study the pharmacologic activity of HU in alleviating NAFLD by measuring blood lipid indicators and morphologic observations of epididymal fat, liver and epididymal fat cells, and lipid deposition in the mouse liver.

## 2. Results and Discussion

### 2.1. Chemical Components of U and HU

Sulfate content of U and HU were 19.9% and 34.7%, respectively (Table 1). Sulfate content of HU was about 1.7 times that of U, demonstrating that HU was successfully synthesized. Total sugar content of HU was 54.1%, higher than that of U. Average molecular weights of HU and U were 22.6 kDa and 143.5 kDa, respectively. Analysis of monosaccharide composition revealed that U and HU were mainly composed of Rhap and Xylp. In addition, both U and HU contained GlcpA, indicating that U and HU were acidic polysaccharides [18].

### 2.2. FT-IR Spectrum Analysis of U and HU

FT-IR analysis revealed the absorptions at 1267 cm^−1^ and 850 cm^−1^ corresponded to the bending variation of S=O and C–O–S, respectively [19,20] (Figure 1). Moreover, the S=O and C–O–S signal intensities of HU were stronger than those of U, which indicates a higher degree of sulfation for HU. The absorption signals of HU and U were approximately the same. This indicates that, except for the increase in sulfate content, no new products were synthesized in the sulfation process. The peak at 918 cm^−1^ may have been caused by β-pyranoside linkage [21].

### 2.3. NMR Analyses of HU and U

The ^1^H NMR (Figure 2A) and COSY NMR (Figure 3A) of U revealed six anomeric proton signals. Their chemical shifts were 4.94, 4.80, 4.95, 4.90, 4.56, and 4.66 ppm, respectively. The peak at 1.36 ppm was due to the C-6 methyl protons of Rhap [22]. There were six anomeric carbon signals at 97.64, 99.99, 98.35, 101.09, 103.10, and 103.34 ppm in ^13^C NMR of U (Figure 2B). The chemical shifts of signals in the range of 60–80 ppm were assigned to sugar ring carbons. Peaks at around 16.96, 57.33, and 175.41 ppm were ascribed to the C-6 of Rhap, carbon atom of -OCH_3_, and the carboxyl signal of uronic acid, respectively [23,24,25].

Through analyzing the COSY and HSQC NMR spectra of U, the main chemical shifts of H and C were observed (Table 2). The signals at 4.94/97.64 and at 4.80/99.99 ppm in the anomeric region of HSQC spectrum were assigned to H1/C1 of residues Ⅰ and Ⅱ, respectively (Figure 3B). The correlated signals of protons and carbon atoms were separately assigned through analyzing the COSY and HSQC spectra (Figure 3A,B). Due to the existence of the sulfate group, H and C exhibited high chemical shifts. Compared with the C-3 (75.50 ppm) in residue Ⅰ, the C-3 (78.32 ppm) shift in residue Ⅱ was higher. Furthermore, the C-4 (78.20 ppm) shift in residue Ⅱ was low-field, such that residue Ⅱ was demonstrated as 1,4-α-L-Rhap3S. By the same way, the low-field chemical shift of C-3 (75.50 ppm) in residue Ⅰ indicated that residue Ⅰ was 1,3-α-L-Rhap [26,27,28].

Residue Ⅲ had an anomeric signal at 4.95/98.35 ppm in the HSQC spectrum (Figure 3B). The shifts of H1–H5 and C1–C6 were obtained from the COSY spectrum and HSQC spectrum, respectively (Figure 3A,B). In addition, a C-6 shift was at around 175.41 ppm, and a C-4 shift at 78.40 ppm was a down-shift, which implied residue Ⅲ was 1,4-β-D-GlcpA [26,28,29].

Residue Ⅳ had the signal at 4.90/101.09 ppm in the anomeric region of the HSQC spectrum of U. According to the COSY spectrum (Figure 3A) and HSQC spectrum (Figure 3B), the chemical shifts of H2-H6 were 4.20, 3.77, 3.67, 3.99, and 3.69 ppm, and the shifts of C2-C6 were 70.00, 74.35, 75.01, 73.48, and 62.67 ppm. Moreover, the shift of C4 was in a low field. Above all, we speculate the residue Ⅳ was 1,4-β-D-Glcp [30].

Both of residue Ⅴ and residue Ⅳ had the same anomeric signal in HSQC spectrum (Figure 3B). Based on the COSY spectrum, the chemical shifts of H1-H6 were 4.90, 4.26, 4.04, 3.69, 3.90, and 3.41 ppm, respectively (Figure 3A). Similarly, the chemical shifts of C1-C6 were 101.90, 68.97, 78.82, 77.01, 76.47, and 62.52 ppm in HSQC spectrum, respectively (Figure 3B). The low-field shift of C-3 (78.82 ppm) demonstrated that -OMe was connected with the C-3 position of residue Ⅴ. As confirmation, in the HMBC spectrum the cross peak at 3.48/78.81 ppm were the corrections between the proton signal (3.47 ppm) of -OMe and C-3 (78.82 ppm) of residue Ⅴ (Figure 3C). In addition, compared with C-5 (76.47 ppm), the C-4 shift was at 77.01 ppm in the low-field region. Based on these results, residue Ⅴ could be identified as 1,4-β-D-Glcp3Me [31].

The signal at 4.56/103.10 ppm in the anomeric region of the HSQC spectrum of U was assigned to residue Ⅵ, and the C-4 shift at 78.38 ppm in the region of the high chemical shift suggested residue Ⅵ was 1,4-β-D-Xylp (Figure 3B) [27,32]. The shifts of proton signals were obtained through analyzing the COSY spectrum, which were 4.56, 3.66, 3.53, 4.16, and 3.41 ppm for H1–H5, respectively (Figure 3A). Similarly, shifts of C1–C5 were identified through the HSQC spectrum (Figure 3B).

Residue Ⅶ had an intense signal at 4.66/103.34 ppm in the anomeric region of the HSQC spectrum of U (Figure 3B). Based on the COSY spectrum, the chemical shifts of H1–H5 were identified (Figure 3A). According to the HSQC spectrum of U (Figure 3B), the carbon atom shifts in residue Ⅶ were assigned. The C-6 shift at 175.41 ppm in residue Ⅶ revealed the presence of uronic acid. In addition, the C-4 shift at 74.00 ppm was at low-field. Therefore, residue Ⅶ could be identified as 1,4-α-L-IdopA [26].

In the ^1^H NMR of HU, the peak at 1.33 ppm was attributed to the C-6 methyl proton in rhamnose (Figure 2C) [22,25]. The strong peak at 4.79 ppm was assigned to the HOD. The signals at 17.21 and 175.22 ppm in the ^13^C NMR of HU were assigned to the C-6 of rhamnose and uronic acid (Figure 2D) [24,25]. In addition, the strong signal at 29.84 ppm was assigned to acetone-d6. Of interest, from the ^13^C NMR, ^1^H NMR, COSY NMR, and HSQC NMR of HU, we found some anomeric signals, and the chemical shifts of H/C were at 5.19/95.99, 4.91/98.13, 4.76/100.71, 5.11/98.13, 4.66/104.03, 4.85/103.10, and 5.15/100.24 ppm. Based on published literature and our results of the NMRs of U, we assigned the anomeric signals at 5.19/95.99, 4.91/98.13, 4.76/100.71, 5.11/98.13, 4.66/104.03, and 4.85/103.10 ppm to sugar residues Ⅰ (1,3-α-L-Rhap), Ⅱ (1,4-α-L-Rhap3S), Ⅲ (1,4-β-D-GlcpA), Ⅳ (1,4-β-D-Glcp), Ⅴ (1,4-β-D-Glcp3Me), and Ⅷ (1,4-α-L-IdopA) in HU [26,27,28,29,31,32]. Chemical shifts of other protons and carbon atoms of sugar residues were identified by analyzing the COSY and HSQC Ns of HU and listed in Table 2.

Compared with chemical shifts of signals in the U NMRs, new signals appeared in the NMRs of HU, which may be the result of the sulfate linkage applied to U. From the COSY and HSQC NMRs of HU, we identified the chemical shifts of protons and carbon atoms for the new signals (Table 2). Sulfate attracts electrons to reduce the electron density around protons and carbon atoms such that the chemical shifts of protons and carbon atoms move to low-field. With this theory in mind, we speculated that the position of the sulfated group was in the sugar residues.

The anomeric signal at 4.66/104.03 ppm in the HSQC NMR of HU was ascribed to the H1/C1 in residue Ⅶ, 1,4-β-D-Xylp2,3S (Figure 4B). From the COSY NMR of HU, the proton signals at 4.27, 4.33, 3.85, 3.89, and 1.33 ppm were assigned to H2-H6, respectively (Figure 4A). By the same way, the shifts of other carbon atoms were identified from the HSQC NMR of HU (Figure 4B). Of interest, compared with the shifts of H/C in U, the H2/C2 (4.22/77.69 ppm) and H3/C3 (4.33/84.74 ppm) shifts of 1,4-β-D-Xylp in HU appeared in the low-field region. Therefore, we speculated that residue Ⅶ was 1,4-β-D-Xylp2,3S.

In the HSQC NMR of HU, the anomeric signal at 5.15/100.24 ppm was assigned to H1/C1 of residue Ⅸ (Figure 4B). The shifts of H1-H6 and C1-C6 were identified by analyzing the COSY and HSQC NMR of HU (Figure 4A,B). Moreover, the H2/C2 (4.67/68.66 ppm) shift of HU was larger than the H2/C2 (4.27/64.38 ppm) shift of U. Based on this and the results of H/C shifts in 1,4-α-L-Rhap3S of U, residue Ⅸ was defined as 1,4-α-L-Rhap2,3S.

Residues Ⅷ and Ⅹ had the same signal (4.85/103.10 ppm) in the anomeric region of the HSQC NMR of HU (Figure 4B). The chemical shifts of protons and carbon atoms were assigned by COSY and HSQC NMRs of HU (Figure 4A,B). Moreover, the shifts of H3/C3 (4.89/72.45 ppm) in residue Ⅹ were higher than the H3/C3 (3.94/70.79 ppm) shift in residue Ⅷ. Thus, residue Ⅹ was assigned as 1,4-α-L-IdopA3S.

The linkage forms of residues in HU were deduced by analyzing the HMBC spectrum of HU (Figure 5). H1 of residue Ⅳ was correlated with C4 of residue Ⅸ (Ⅳ H1/Ⅸ C4), which confirmed residue Ⅳ was attached at the 4-position of residue Ⅸ. The correlations between H1 of residue Ⅱ and C4 of residue Ⅶ (Ⅱ H1/Ⅶ C4) showed that there was a linkage between residue Ⅱ and the C4 of residue Ⅶ. By the same way, residue Ⅹ had interactions between H1 and the C4 of residue Ⅱ (Ⅹ H1/Ⅱ C4), indicating that residue Ⅹ was probably attached at the 4-position of residue Ⅱ. There were also correlations between H1 of residue Ⅲ and C3 of residue Ⅰ (Ⅲ H1/Ⅰ C3), demonstrating that the 1-position of residue Ⅲ was attached at the C3 of residue Ⅰ. In addition, residue Ⅴ had interactions between H4 and C1 of residue Ⅹ (Ⅴ H4/Ⅹ C1), confirming that there was bond between residue Ⅴ and the C1 of residue Ⅹ. Residue Ⅹ had interactions from H4 to C1 of residue Ⅶ (Ⅹ H4/Ⅶ C1), implying that there was linkage from the 1-position of residue Ⅶ to the C4 of residue Ⅹ.

These results indicate that the plausible structural fragments of HU are as follows: →4)-β-D-Glcp(1→4)-α-L-Rhap2,3S(1→;→4)-α-L-Rhap3S(1→4)-β-D-Xylp2,3S(1→;→4)-α-L-Rhap3S(1→4)-β-D-Xylp (1→;→4)-α-L-IdopA3S(1→4)-α-L-Rhap3S(1→;→4)-β-D-GlcpA(1→3)-α-L-Rhap(1→;→4)-α-L-IdopA3S(1→4)-β-D-Glcp3Me(1→;→4)-β-D-Xylp2,3S(1→4)-α-L-IdopA3S(1→; and→4)-β-D-Xylp(1→4)-α-L-IdopA3S(1→ (Figure 6).

OSO_3_: both positions 2 and 3 of xylose are substituted by sulphate, or one or neither position is substituted.

### 2.4. Effect of HU on Body Weight in Mice with NAFLD

A sedentary lifestyle and high fat eating habits are considered direct contributors to the development of NAFLD [33]. To examine the effects of HU on NAFLD, male C57BL/6 mice were fed either an HFD or normal diet for 16 weeks and treated with or without polysaccharides for 14 weeks. There was no significant difference in initial animal weight among the five groups (*p* > 0.05). As the study progressed, mice in the NAFLD group showed increasing slow activity and poor response. Mice in all groups gained weight, with the MOD group showing the most weight gain (Figure 7A). However, after administration of U and HU, weight gain was suppressed. There was a significant difference in weight between the HU and MOD groups (*p* < 0.01).

### 2.5. Effect of HU on the Abnormal Changes in Lipid Metabolism Indicators Caused by NAFLD

NAFLD is a hepatic manifestation of a metabolic syndrome and is accompanied by abnormal changes in blood lipids. Compared to mice fed a standard diet, the serum TG level in mice fed an HFD significantly increased (*p* < 0.01) and even reached a high of 0.72 ± 0.06 mmol/L (Figure 7B). Following HU treatment, TG decreased by 45.7% (*p* < 0.01), dropping to 0.39 ± 0.07 mmol/L. This result was similar to the NOR group (0.38 ± 0.02 mmol/L) and the Feno group (0.37 ± 0.02 mmol/L).

Excessive fat intake also leads to dyslipidemia. In this study, the level of LDL-C in the MOD mice reached a high of 0.30 ± 0.03 mmol/L. After administration of HU and U, LDL-C levels were reduced by 30.2% in the HU group (0.21 ± 0.01 mmol/L) and by 21.4% in the U group (0.24 ± 0.01 mmol/L). In addition, the inhibitory effects of both HU and U on the increase in LDL-C level were significant (*p* < 0.01) (Figure 7D). Furthermore, the ratio of HDL-C/LDL-C indicated that HU (*p* < 0.05) appeared to attenuate dyslipidemia changes caused by NAFLD more so than U (*p* > 0.05) when compared with the MOD group (Figure 7E).

The liver is the primary organ of cholesterol metabolism. Disturbance in the metabolism balance in the liver can lead to accumulation of lipids in the organ and to development of NAFLD. In this study, compared with the MOD group, TC levels significantly decreased from 4.59 ± 0.07 mmol/L to 3.95 ± 0.17 and 4.18 ± 0.05 mmol/L after HU and U treatments, respectively, levels of which were lower than the Feno group (4.33 ± 0.06 mmol/L). Statistical analysis showed that HU had a better trend of action (compared with the MOD group *p* < 0.01), although U attenuated elevated serum TC level associated with NAFLD as well (compared with the MOD group *p* < 0.05) (Figure 7C).

Inhibiting the reabsorption and increasing the excretion of bile acids (BAs) may be a main pathway for HU and U to exert a protective effect against NAFLD. BAs as cholesterol metabolites are involved in lipid metabolism and are released into the intestine to participate in lipid transport [34]. BAs facilitate digestion and absorption of nutrients, after which 95% of the BAs are reabsorbed by the intestine and transported back to the liver, and the rest are excreted with feces [35]. Research has shown that polysaccharides derived from some marine sources affect BA metabolism by reducing reabsorption of BAs, promoting the conversion from cholesterol to bile acid, and decreasing the level of lipids in the liver. For example, Ma et al. [36] found that oyster polysaccharide decreases expression of farnesoid X receptor (FXR) in the gut to reduce reabsorption of BAs. Gao et al. [37] showed that low molecular weight *Laminaria japonica* polysaccharide has a high bile acid binding capacity, which increases excretion of BAs. Our early investigation on *U pertusa* found that fecal BA content of hypercholesterolemic rats increased significantly after treatment with U [5].

### 2.6. Effect of HU on Liver Function in NAFLD Mice

In the MOD group, serum AST was 165.00 ± 7.492 U/L, and compared with the NOR group, it was increased by 64.18%, indicating that hepatocytes in the MOD mice were destroyed (Figure 7F). AST in the HU group (114.0 ± 10.98 U/L) decreased by 30.9% compared with the MOD group (*p* < 0.01), which was similar to the Feno group (113.5 ± 3.11 U/L). After treatment with U, the AST level in the NAFLD mice was reduced to 155.2 U/L, but the reduction was not significant.

ALT is mainly present in the cytoplasm of hepatocytes, and the AST/ALT level is positively correlated with the severity of hepatocyte necrosis. After administration of HU, not only was the AST level significantly reduced, but the AST/ALT ratio compared with the MOD group was also lower than that of U (*p* < 0.01). The results showed that HU has an inhibitory effect on ameliorating the abnormal changes in liver function induced by NAFLD compared with U (Figure 7G), although treatment with U and HU both decreased the ASL/ALT levels in NAFLD mice. Reduction in these blood parameters appears to indicate that damage to the liver was reduced following HU administration.

### 2.7. Effect of HU on Epididymal Adipose Tissue Changes Associated with NAFLD

TG is synthesized by the liver and stored in adipose cells. HFD-induced enlarged epididymal adipose cells affect epididymal fat volume [38]. In this study, adipose tissue volume was the largest in the MOD group. Following administration of U and HU, adipose tissue volume was reduced (Figure 8A). In addition, H&E staining revealed that the MOD group had the fewest number of adipose cells compared with other groups (Figure 8B,C). This was confirmed through semiquantitative analysis of H&E staining of the adipose cells. Of interest, the number of cells in the HU group was larger than that in the U group (*p* < 0.001), which suggests that the treatment effect of HU was better than U (Figure 8H).

### 2.8. Effect of HU on Liver Damage Caused by NAFLD

Morphologically, U and HU administration led to a tighter and more ruddy liver appearance, whereas the livers in the MOD mice fed HFD were paler, and their surfaces had an oily film (Figure 8D).

H&E staining revealed that hepatocyte morphology in the NOR group was normal with abundant cytoplasm and a round central nucleus, but in the MOD group, the nucleus shifted due to the large amount of lipid droplets that occupied the cells (Figure 8E,F). H&E staining results assessed by semiquantitative analysis found that, compared with the MOD group, after HU treatment, the vacuole area was significantly reduced (*p* < 0.01), and the extent of liver damage was reduced (Figure 8I). Furthermore, results from oil red O staining showed that lipid accumulation was alleviated in hepatocytes (Figure 8G). Although U also relieved lipid deposition in the liver of NAFLD mice, the therapeutic effect of HU was significantly better than U (Figure 8J) (*p* < 0.01 vs. U group).

Studies have demonstrated that introduction of a sulfate group enhances the negative charge of polysaccharides and intramolecular repulsion [39,40]. Sulfation also alters molecular chain conformation and flexibility in polysaccharides to expose more biologically active units, further augmenting the biologic activities of polysaccharides [41,42], which may account for the heightened efficacy of HU over U in our study. He et al. found *Sarcodia ceylonensis* polysaccharide (SCP), *Ulva lactuca L.* polysaccharide(ULLP), and *Durvillaea antarctica* polysaccharide (DAP) were all sulfated polysaccharides, except for *Gracilaria lemaneiformis* polysaccharide (GLP). Moreover, the antioxidant activity of SCP, ULLP and DAP were higher than that of GLP [43]. Similarly, the serum LDL-C levels and AST concentrations were lower in type 2 diabetic rats treated with high sulfate content *Macrocystis pyrifera* (L.) *Ag*. polysaccharides [44]. Changes in molecular weight also have effects on the biological activity of polysaccharides. Low-molecular weight fucoidan inhibits the differentiation of osteoclasts and reduces osteoporosis in ovariectomized rats [45]. Low molecular weight fucoidan (LMWF), a sulfated polysaccharide extracted from brown seaweeds, has shown strong anti-inflammatory and antioxidant activities. LMWF prevents NAFLD in db/db mice by activation of the SIRT1/AMPK/PGC1α signaling pathway [46]. In our work, HU has a lower molecular weight than U, and the anti-NAFLD effect of HU appears to be stronger in reducing the serum concentration of TC, TG, AST, and LDL-C.

The gut microbiota can be compared to a neglected organ and a second genome [47,48], while marine algae polysaccharides are considered a promising prebiotic for the health of an organism [49,50,51]. *Sargassum fusiforme* fucoidan re-established the gut microbiota, including reducing the number of harmful bacteria *Helicobacter* and increasing the number of beneficial bacteria *Bacteroides*, *Lactobacillus*, and *Alistipes*. This prebiotic effect of fucoidan may be responsible for the reduction of insulin resistance, the improvement of glucose and lipid metabolism disorders, and the enhancement of intestinal barrier function [52]. Neoagarooligosaccharides from *Gelidium elegans* increase the abundance of gut microbes such as *Eubacterium fissicatena* and *Ruminococcaceae UCG-005* [53]. These bacteria were negatively associated with body weight and obesity-related metabolic factors, suggesting that neoagarooligosaccharides target the gut microbiota to improve obesity [54,55]. In addition, *Ulva prolifera* polysaccharide supplementation not only increased the antiobesity in bacteria such as *Parasutterella*, *Faecalibaculum*, and *Bifidobacteria* but also inhibited microorganisms associated with inflammation and metabolic disorders including *Acetatifactor*, *Tyzzerella*, *Desulfovibrio*, and *Ruminoccoccus_1* [56].

The gut–liver axis not only involves the above-mentioned absorption and excretion of the BAs but also the production and utilization of short-chain fatty acids (SCFAs) [57]. SCFAs are produced when gut bacteria ferment dietary fibers and resistant starch. SCFAs are known to help reduce inflammation and support metabolism, among other health benefits [58]. The type and amount of SCFAs synthesized in the gut change due to dysbiosis, which may lead to NAFLD [59,60]. Seaweed polysaccharides are not easily digested and are fermented by intestinal bacteria to produce SCFAs [61,62]. The three main types of SCFAs are acetate, propionate, and butyrate. Acetate and propionate are involved in liver lipogenesis and gluconeogenesis, respectively. In particular, acetate is a precursor to cholesterol or fatty acids [63]. Studies on mice fed an HFD showed that SCFAs reduced liver fatty acid synthase activity and liver lipid synthesis [64].

From the results of our study, we speculate that the effect of HU in treating NAFLD was through the gut–liver axis. In future laboratory and animal studies, we will attempt to elucidate the mechanism of HU in treating NAFLD by detecting levels of bile acids in serum and fecal material and assessing the types and content of SCFAs after fermentation of HU with gut microbiota.

## 3. Materials and Methods

### 3.1. Materials and Reagents

Fresh *U pertusa* was obtained from the Qingdao coastal area of China. The algae species was identified by Chongmei Xu (Associate Professor, Weifang Medical University, China). Dialysis membrane with 3500 Da cutoff was purchased from Spectrum Chemical Manufacturing (New Brunswick, NJ, USA). All other chemical reagents were analytical grade.

### 3.2. Isolation of U

U was isolated from *U pertusa* using the method described by Yu et al. [5]. Briefly, dry *U pertusa* immersed in tap water was boiled for 4 h. The water extract was collected and concentrated to 25% of the initial volume under vacuum at 60 °C. The concentrated water extract was added to a mixture of ethanol and KCl to form final concentrations of 75% and 0.3%, respectively. The mixture was allowed to stand for 24 h to precipitate the polysaccharide. Finally, after soaking the precipitate in absolute ethanol for 36 h and drying at 60 °C, U was obtained.

### 3.3. Preparation of HU

HU was prepared by derivatization of U with a sulfation reagent [17]. The sulfation reagent was prepared by combining chlorosulfonic acid and formamide (v:v = 1:6) and reacted in an ice bath. U (4.0 g) was added to formamide (160 mL) and mixed for 30 min at 60 °C, after which the sulfation reagent (50 mL) was added. After reacting for 4 h, the mixture was cooled to room temperature and neutralized with 30% NaOH. Then, the mixture was dialyzed with tap water for 24 h and distilled water for 48 h. Finally, after concentrating and freeze-drying, HU was prepared.

### 3.4. Chemical Analysis of U and HU

Total sugar content was measured using the phenol-sulfuric acid method [65]. Rhamnose was used as a standard to detect the absorbance of the polysaccharide sample at 490 nm to obtain the total sugar content. Sulfate content was detected through the barium chloride-gelatin method [66]. Briefly, we added the HCl solution (0.2 mol/L) and the configured BaCl_2_-gelatin solution to the polysaccharide solution, then mixed well and let stand for 20 min. Next, we measured the OD value at 500 nm. In this experiment, K_2_SO_4_ was used as the standard.

### 3.5. Determination of Molecular Weights of U and HU

Molecular weights of U and HU were determined using high-performance liquid chromatography (HPLC). Briefly, the molecular weights of U and HU were detected by a TSK-Gel G3000PWXL column (7.5 × 300 mm; Tosoh, Tokyo, Japan) using HPLC (LC-20A; Shimadzu, Kyoto, Japan) under the following experimental conditions: column temperature 40 °C, 0.1 mol/L Na_2_SO_4_ as the mobile phase, and a flow rate of 0.5 mL/min. Dextran was used as the standard.

### 3.6. Analysis of Monosaccharide Compositions of U and HU

The PMP precolumn derivatizations of U and HU were performed using the flow the method of Wen, Cui, Dong, and Zhang [67]. The monosaccharide compositions of U and HU were measured using the HPLC YMC-Pack ODS-AQ column (250 × 4.6 nm, 5 μm; LC-20 A; Shimadzu) under the following conditions: column temperature 30 °C, the mixture of 0.4% triethylamine and acetonitrile (9:1) as mobile phase A, the mixture of 0.4% triethylamine and acetonitrile (4:6) as mobile phase B, and flow rate of 1.0 mL/min. In addition, rhamnose (Rhap), xylose (Xylp), mannose (Manp), glucose (Glcp), glucuronic acid (GlcpA), and fucose (Fucp) were used as the monosaccharide standards.

### 3.7. Analysis of FT-IR and NMR Spectrum of U and HU

The dried samples were crushed with KBr powder using a mortar and pestle, and the resulting powder was pressed into pellets, which were then transferred to a Thermo/Nicolet Magna-Avatar 360 spectrometer (Thermo Scientific; Waltham, MA, USA) to obtain the FT-IR spectra. U was dissolved in D_2_O solvent, and HU was dissolved in a mixed solvent (D_2_O+C_3_D_6_O). The ^1^H-NMR, ^13^C-NMR, and 2D NMR (COSY, HSQC, HMBC) spectra of U and HU were obtained using an ultra-low temperature probe of a 600 MHz NMR spectrometer (600 MHz; AVANCE III; Bruker; Billerica, MA, USA).

### 3.8. Animal Models and Experimental Design

Male C57BL/6J mice were used as the NAFLD model, which was constructed to verify the effects of HU on NAFLD induced by a high-fat diet (HFD). All mice were eight weeks old when purchased from Vital River Laboratory Animal Technology (Beijing, China). They were housed in the animal barrier system at the Animal Laboratory Center of Weifang Medical University, where the temperature was maintained at 22 ± 2 °C with 50% ± 10% relative humidity, and a 12:12 h light/dark cycle.

After 1 week of adaptation, thirty male mice weighing 22–25 g were randomly divided into five groups with six mice in each group. The groups were designated normal diet (NOR), model (MOD), positive drug, U, and HU. The positive drug group mice were given fenofibrate solution at an intragastric dose of 30 mg/kg/d. The NOR and MOD mice were each given an equal volume of 0.2 mL distilled water intragastrically. Mice in the remaining two groups were given orally U or HU at 250 mg/kg/d. Except for the NOR group, mice in the other four groups were fed with HFD purchased from Huafukang Biotechnology (Beijing, China) for 16 continuous weeks. For the first two weeks, the animals only received feedings; then the gavage experiment was performed from the third through the sixteenth weeks [68]. Body weights were recorded once weekly, and the mental state of the mice was observed throughout the study. At the end of the experiment (week 16), all animals were weighed and anesthetized with 1% (*w/v*) sodium pentobarbital after overnight fasting. Blood samples were collected, and fresh mouse liver as well as epididymal fat tissue were harvested and washed with saline.

Animal experiments aimed to minimize distress to the animals and were performed according to the National Institutes of Health *Guide for the Care and Use of Laboratory Animals* (2011) and were approved by the Weifang Medical University Ethics Committee.

### 3.9. Biochemical Analysis of Blood Samples

Blood samples were collected from anesthetized mice via retro-orbital sampling. Blood sample tubes were allowed to stand at room temperature for 60 min followed by centrifugation for 15 min at 860 g. All serum biochemical indicators were measured using a biochemistry analyzer (Cobas 8000; Roche Diagnostics, Basel, Switzerland). The serum biochemistry indicators included total cholesterol (TC), triglycerides (TG), high-density lipoprotein cholesterol (HDL-C), low-density lipoprotein cholesterol (LDL-C), the ratio of HDL-C and LDL-C, glutamic oxaloacetic transaminase (AST), glutamic pyruvic transaminase (ALT), and the AST:ALT ratio.

### 3.10. Histopathologic Observations

Harvested livers were soaked overnight in 4% paraformaldehyde solution. Liver tissues then underwent paraffin embedding, continuous sectioning, dewaxing, hematoxylin–eosin (H&E) staining, and neutral gum sealing, among other testing. The liver tissues were also stained with oil red O. Epididymal adipose tissue was immersed in fixation fluid, followed by the H&E staining and processing as described above. Histomorphologic changes in the liver and epididymal fat tissue of each group were observed under 200× and 400× light microscopy (BX43; Olympus; Tokyo, Japan). Areas of liver lipid droplets and number of adipose tissue cells underwent semiquantitative computational evaluation using ImageJ software (National Institutes of Health, Bethesda, MD, USA). The percentage of liver parenchyma occupied by fatty liver cells was divided into four levels: 0–5%, 5–33%, 33–66%, or more than 66% [69].

### 3.11. Statistical Analysis

All data were expressed as mean ± standard error of mean (SEM). Statistical comparisons of data were carried out using one-way ANOVA by GraphPad Prism software version 6.0 (GraphPad Software Inc., La Jolla, CA, USA). Significant differences were set at *p* < 0.05.

## 4. Conclusions

We evaluated the main structural fragments and anti-NAFLD effect of HU. The results indicated that the main structural fragments of HU were →4)-β-D-Glcp(1→4)-α-L-Rhap2,3S(1→; →4)-α-L-Rhap3S(1→4)-β-D-Xylp2,3S(1→; →4)-α-L-Rhap3S(1→4)-β-D-Xylp(1→; →4)-α-L-IdopA3S(1→4)-α-L-Rhap3S(1→; →4)-β-D-GlcpA(1→3)-α-L-Rhap(1→; →4)-α-L-IdopA3S(1→4)-β-D-Glcp3Me(1→; →4)-β-D-Xylp2,3S(1→4)-α-L-IdopA3S(1→; and →4)-β-D-Xylp(1→4)-α-L-IdopA3S(1→. In NAFLD mice, HU not only improved serum lipid indicators (TC, TG, LDL-C, LDL-C/HDL-C) and liver function parameters (AST, AST/ALT) but also inhibited the abnormal enlargement of epididymal fat cells and massive deposition of lipid in the liver. Our current study highlights the potential of HU supplement as a functional food for preventing NAFLD. As a next step, we will explore HU in terms of dose dependence and toxicology as an anti-NAFLD drug. In addition, we plan to study the role and mechanism of HU in the treatment of NAFLD using animal experiments with fecal bacteria transplantation.

## Figures and Tables

**Figure 1 pharmaceuticals-16-00062-f001:**
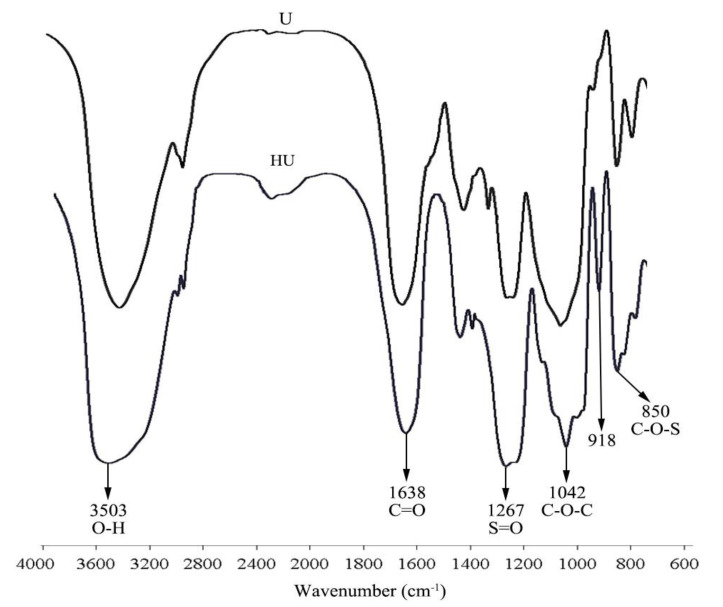
FT-IR spectrum of U and HU.

**Figure 2 pharmaceuticals-16-00062-f002:**
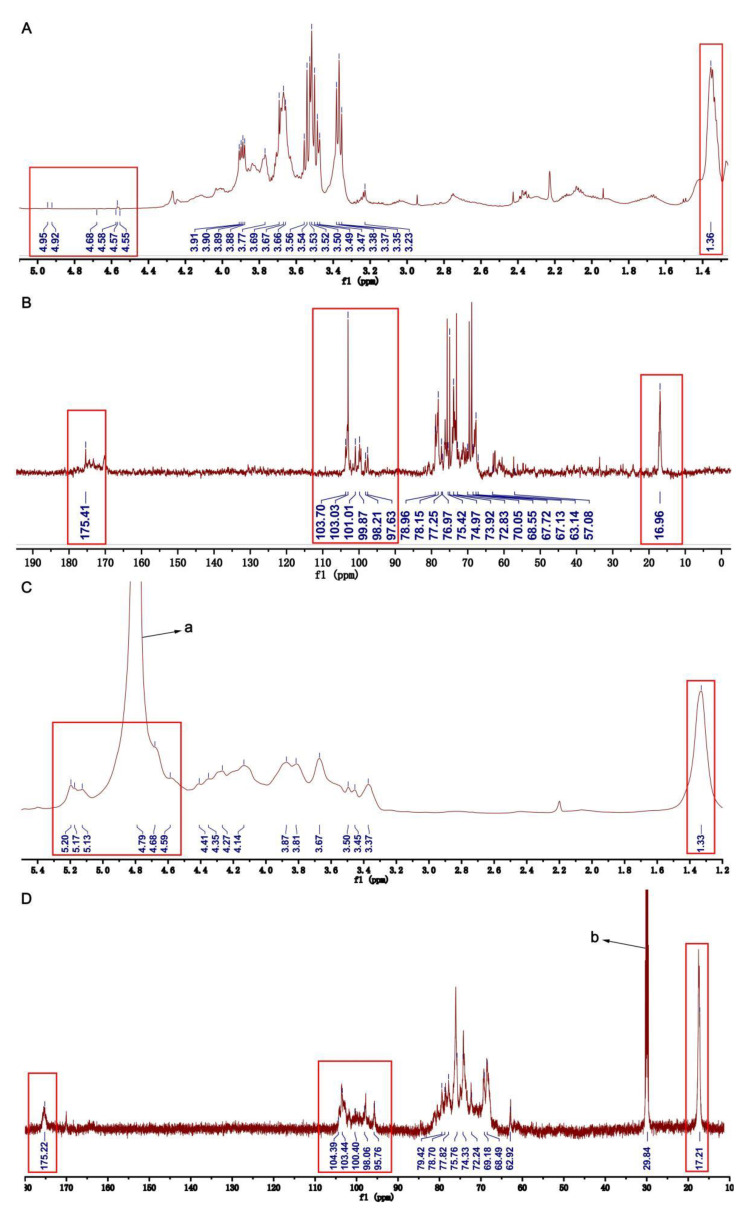
One-dimensional NMR of U and HU. (**A**) ^1^H NMR of U. (**B**) ^13^C NMR of U. (**C**) ^1^H NMR of HU. (**D**) ^13^C NMR of HU. (**a**) The peak of HOD. (**b**) The peak of acetone-d6 (defining peak).

**Figure 3 pharmaceuticals-16-00062-f003:**
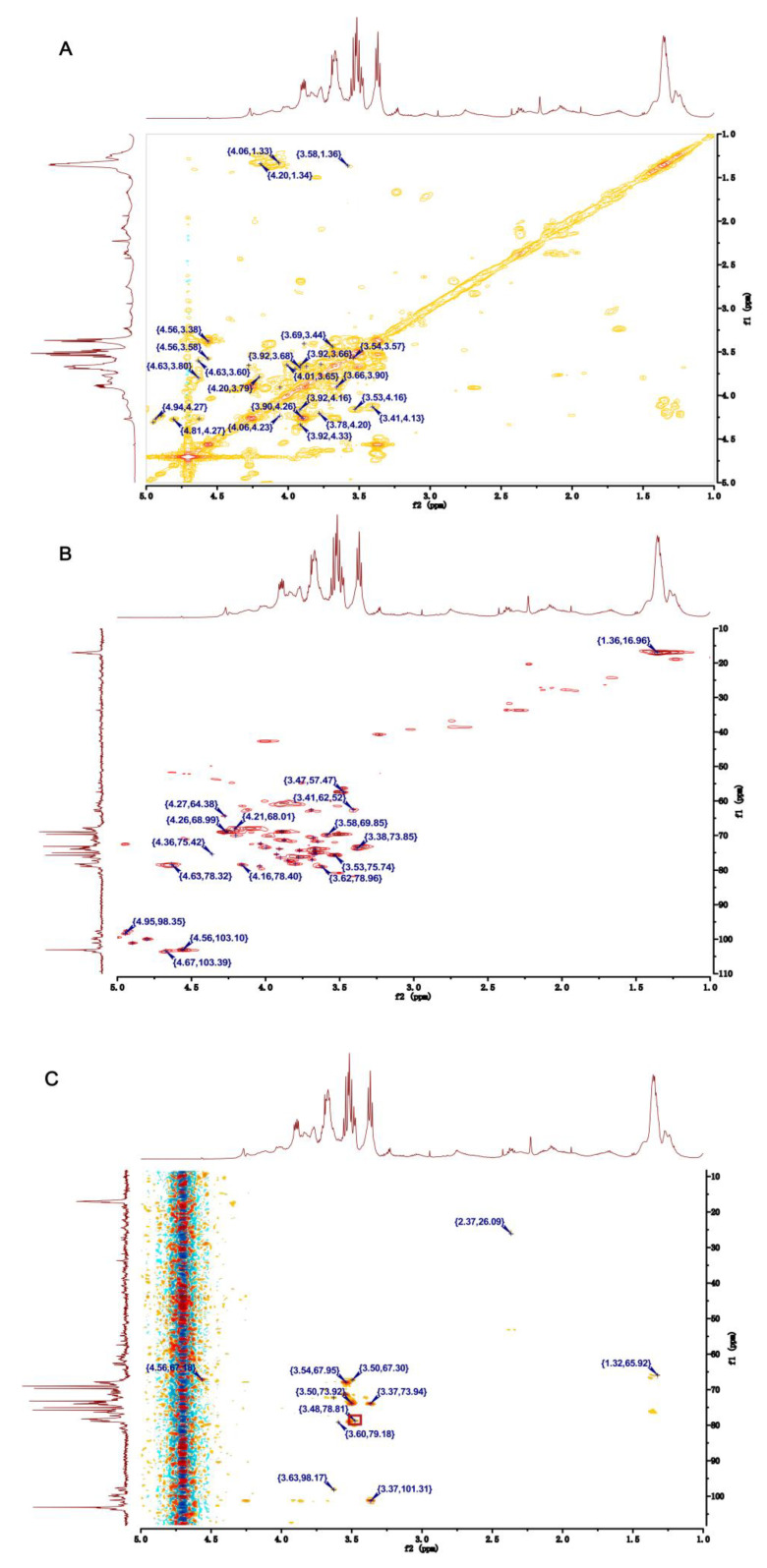
Two-dimensional NMR of U. (**A**) COSY NMR of U. (**B**) HSQC NMR of U. (**C**) HMBC NMR of U.

**Figure 4 pharmaceuticals-16-00062-f004:**
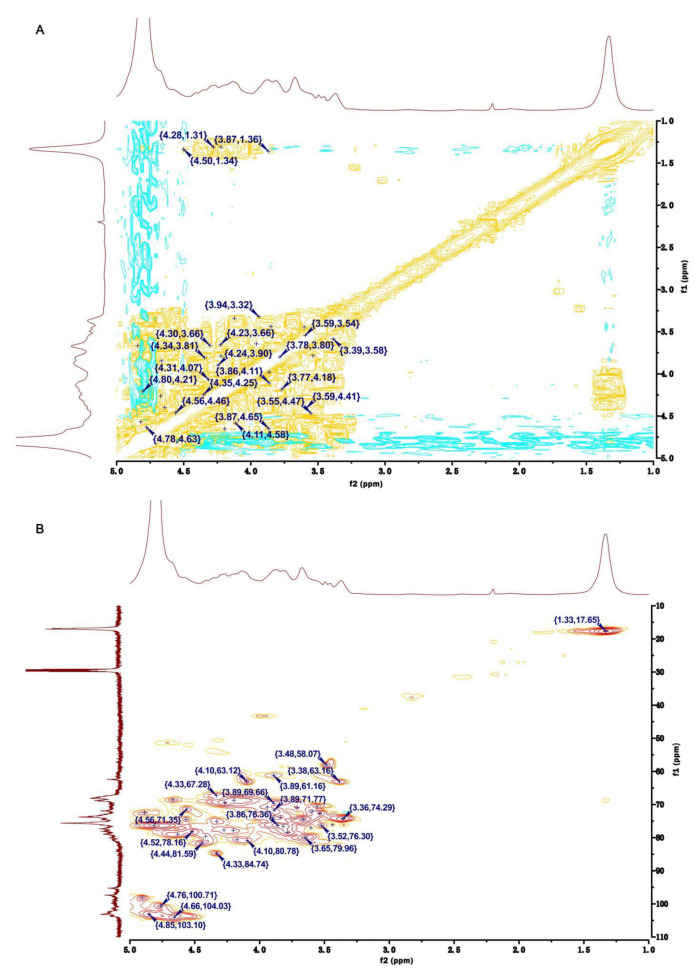
Two-dimensional NMR of HU. (**A**) COSY NMR of HU. (**B**) HSQC NMR of HU.

**Figure 5 pharmaceuticals-16-00062-f005:**
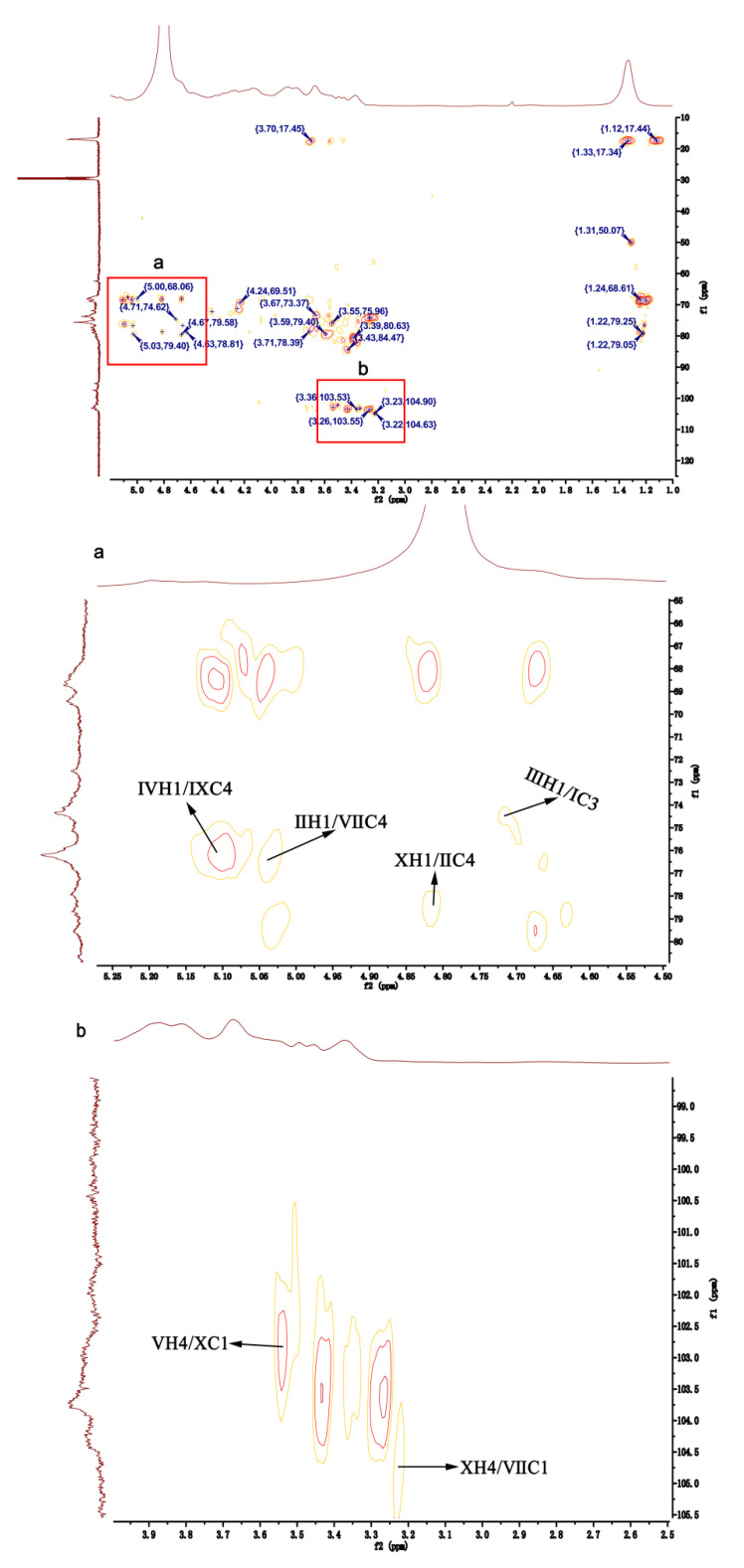
HMBC NMR of HU.

**Figure 6 pharmaceuticals-16-00062-f006:**
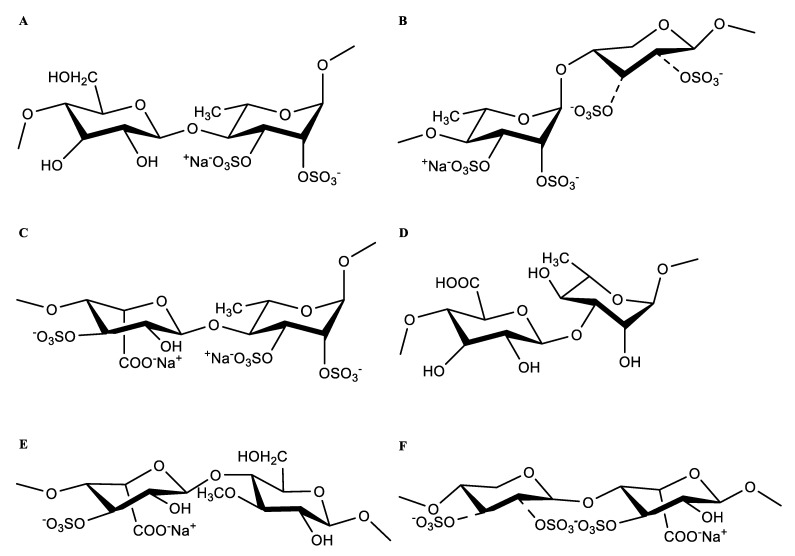
The main form of residues linkage of HU. (**A**) →4)-β-D-Glcp(1→4)-α-L-Rhap2,3S(1→. (**B**) →4)-α-L-Rhap3S(1→4)-β-D-Xylp2,3S(1→. (**C**) →4)-α-L-IdopA3S(1→4)-α-L-Rhap3S(1→. (**D**) →4)-β-D-GlcpA(1→3)-α-L-Rhap(1→. (**E**) →4)-α-L-IdopA3S(1→4)-β-D-Glcp3Me(1→. (**F**) →4)-β-D-Xylp2,3S(1→4)-α-L-IdopA3S(1→.

**Figure 7 pharmaceuticals-16-00062-f007:**
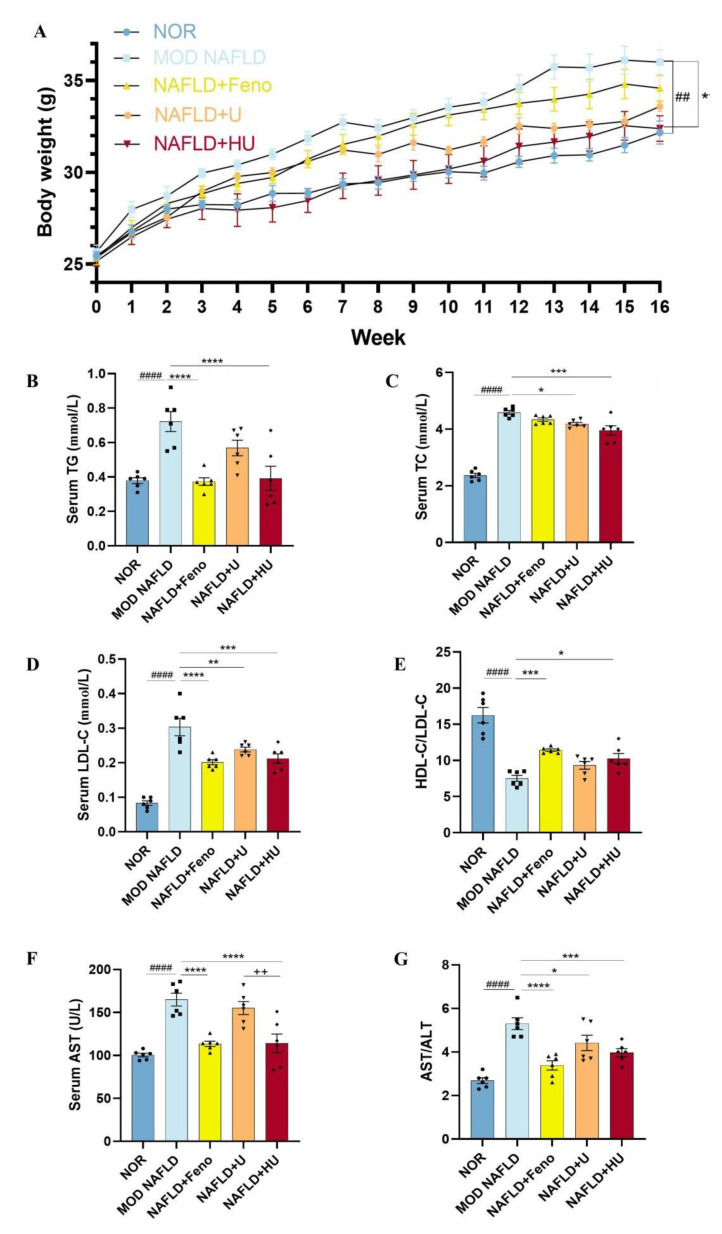
Effects of U and HU on body weight, lipid metabolism indicators, and liver function in NAFLD mice. (**A**) Weight change over time. Results of blood parameters: (**B**) TG, (**C**) TC, (**D**) LDL-C, (**E**) HDL-C and LDL-C ratio, (**F**) ALT, (**G**) AST and ALT ratio. Data are shown as the mean ± SEM. ^##^ *p* < 0.01, ^####^ *p* < 0.01 vs. NOR group; * *p* < 0.05, ** *p* < 0.01, *** *p* < 0.005, **** *p* < 0.0001 vs. MOD group; ^++^ *p* < 0.01 vs. U group.

**Figure 8 pharmaceuticals-16-00062-f008:**
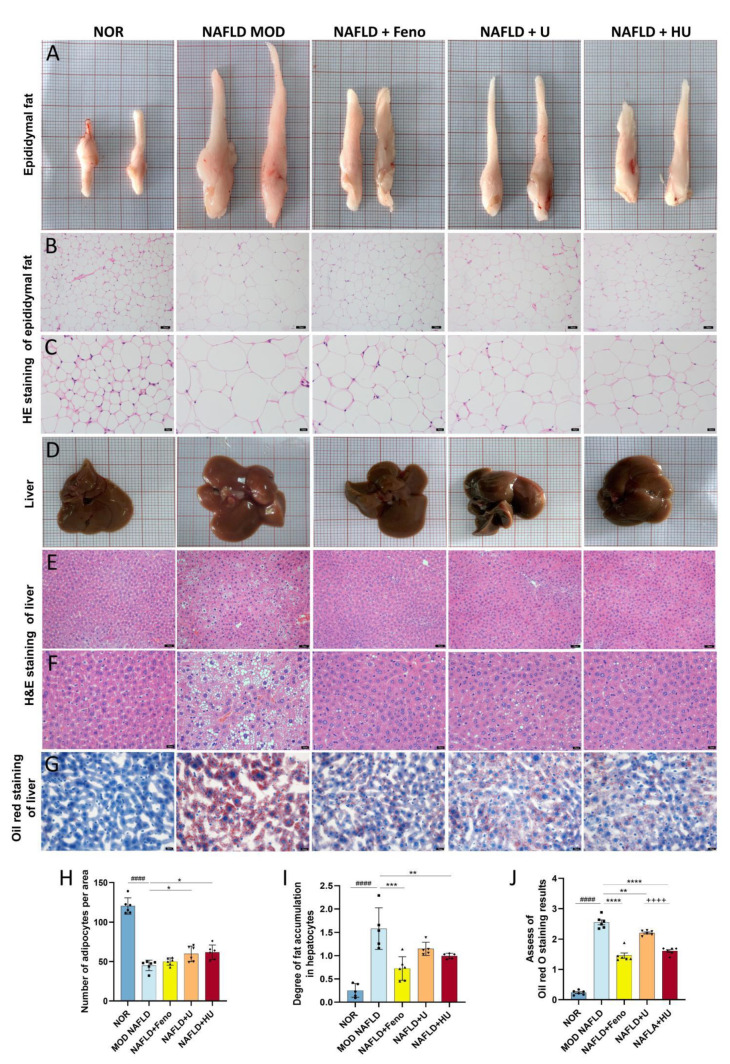
Effects of U and HU on epididymal fat and liver in NAFLD mice. (**A**) Epididymal fat morphology. Epididymal fat H&E staining magnification 200× (scale bar = 50 µm) (**B**) and 400× (scale bar = 20 µm) (**C**). (**D**) Morphology of liver tissue. H&E stained liver tissue, magnification 200× (scale bar = 50 µm) (**E**) and 400× (scale bar = 20 µm) (**F**). (**G**) Oil red stained liver tissue, magnification 400× (scale bar = 20 µm). (**H**) Semiquantitative assessment of H&E stained epididymis adipose tissue. (**I**) Semiquantitative assessment of H&E stained liver tissue. (**J**) Semiquantitative assessment of oil red stained liver tissue. Data are shown as mean ± SEM. ^####^ *p* < 0.01 vs. the NOR group; * *p* < 0.05, ** *p* < 0.01, *** *p* < 0.005, **** *p* < 0.0001 vs. MOD group; ^++++^ *p* < 0.0001 vs. U group.

**Table 1 pharmaceuticals-16-00062-t001:** Chemical components of U and HU.

Scheme	Total Sugar (%)	Sulfate (%)	Average Molecular Weights (kDa)	Monosaccharide Components (mol%)					
				Rhap	Xylp	Glcp	GlcpA	Fucp	Manp
U	45.2	19.9	143.5	31.3	19.9	6.7	5.5	0.5	--
HU	54.1	34.7	22.6	30.7	18.1	6.5	5.4	0.6	0.5

Note: “--”, not detected.

**Table 2 pharmaceuticals-16-00062-t002:** ^1^H and ^3^C NMR chemical shifts of U and HU recorded in D_2_O.

Polysaccharides	Name	Sugar Residues	Chemical Shifts (ppm)						
			H1/C1	H2/C2	H3/C3	H4/C4	H5/C5	H6/C6	-OMe
U	Ⅰ	-3)-α-L-Rhap(1-	4.94/97.64	4.26/68.99	3.93/75.50	3.68/70.72	4.03/72.41	1.36/16.96	
	Ⅱ	-4)-α-L-Rhap3S(1-	4.80/99.99	4.27/64.38	4.63/78.32	3.80/78.20	4.21/68.01	1.34/16.96	
	Ⅲ	-4)-β-D-GlcpA(1-	4.95/98.35	4.36/75.42	3.91/73.92	4.16/78.40	3.58/69.85	--/175.41	
	Ⅳ	-4)-β-D-Glcp(1-	4.90/101.09	4.20/70.00	3.77/74.35	3.67/75.01	3.99/73.48	3.69/62.67	
	Ⅴ	-4)-β-D-Glcp3Me(1-	4.90/101.09	4.26/68.97	4.04/78.82	3.69/77.01	3.90/76.47	3.41/62.52	3.47/57.33
	Ⅵ	-4)-β-D-Xylp(1-	4.56/103.10	3.66/71.77	3.53/75.74	4.16/78.38	3.41/62.52	--/--	
	Ⅶ	-4)-α-L-IdopA(1-	4.66/103.34	3.38/73.85	3.87/71.37	3.67/74.00	4.27/64.40	--/175.41	
HU	Ⅰ	-3)-α-L-Rhap(1-	5.19/95.99	4.19/69.73	3.78/75.34	3.84/73.86	3.53/72.70	1.33/17.21	
	Ⅱ	-4)-α-L-Rhap3S(1-	4.91/98.13	4.07/66.53	4.33/75.08	3.65/79.96	4.26/69.66	1.33/17.21	
	Ⅲ	-4)-β-D-GlcpA(1-	4.76/100.71	4.63/78.97	3.89/71.77	4.10/80.78	4.52/78.16	--/175.22	
	Ⅳ	-4)-β-D-Glcp(1-	5.11/98.13	4.63/79.02	4.42/79.58	3.60/77.00	3.44/76.09	3.89/61.16	
	Ⅴ	-4)-β-D-Glcp3Me(1-	5.11/98.13	4.57/74.46	4.44/81.59	3.52/76.32	3.59/72.09	3.38/63.16	3.48/58.07
	Ⅵ	-4)-β-D-Xylp(1-	4.66/104.03	3.89/73.95	4.20/77.80	3.80/76.58	3.71/70.99	--/--	
	Ⅶ	-4)-β-D-Xylp2,3S(1-	4.66/104.03	4.27/77.69	4.33/84.74	3.85/76.36	3.89/69.66	--/--	
	Ⅷ	-4)-α-L-IdopA(1-	4.85/103.10	3.66/74.59	3.94/70.79	3.36/74.17	4.10/63.12	--/175.22	
	Ⅸ	-4)-α-L-Rhap2*,*3S(1-	5.15/100.24	4.67/68.66	4.17/80.62	4.81/76.29	4.56/71.35	1.33/17.21	
	Ⅹ	-4)-α-L-IdopA3S(1-	4.85/103.10	3.66/74.59	4.89/72.45	3.36/74.17	4.10/63.12	--/175.22	

Note: “--”, not detected.

## Data Availability

Data is contained within article.

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
