# Peer review of "Structural Characterization and Anti-Nonalcoholic Fatty Liver Effect of High-Sulfated Ulva pertusa Polysaccharide"

_pharmaceuticals, 2022, doi:10.3390/ph16010062_

Round 1

Reviewer 1 Report

In the manuscript, “Structural characterization and anti-nonalcoholic fatty liver effect of high-sulfated Ulva pertusa polysaccharide”, the author explored the main structure of HU and its therapeutic effect against nonalcoholic fatty liver disease (NAFLD). The main structure of HU was elucidated using FT-IR and NMR (13C, 1H, COSY, HSQC, HMBC). The anti-NAFLD activity of HU was explored using the high-fat diet mouse model to detect indicators of blood lipid and liver function and observe the pathologic changes in epididymal fat and the liver. And this manuscript has clear logic and smooth sentences, but it also has some places that need revision. Comments are as follows:

1.    What is “197” mean in line 167? The title of the introduction had a format mistake.

2.    The author mentioned HU plays a role in improving liver damage caused by NAFLD through the gut-liver axis, there also has a theory that polysaccharides balance the gut microbe to keep host health, is suggest to disccuss this topic in the MS, and some some relevant literatures are recommend to discuss, e.g. https://doi.org/10.3390/foods11223550; and https://dx.doi.org/10.1021/acs.jafc.0c02555

3.    Why use epididymal adipose tissue to evaluate TG?

4.    The author had done the characterization analysis of HU, so the author could discuss more mechanisms of liver protection.

5.    The conclusion part is not simply to repeat the results, it is better to indicate the potential future research possibilities or application and show the importance and advantage of your study.

Author Response

请参阅附件

Reviewer 2 Report

This is a wonderful manuscript. The authors explored the main structure of high-sulfated ulvan (HU) and its therapeutic effect against nonalcoholic fatty liver disease (NAFLD). In addition, they also discussed why sulfation enhanced the biological activity of polysaccharides. In conclusion, this manuscript suggests that HU may be a potential treatment for NAFLD. However, the following are some suggestions that have to be considered to make this article great.

1. Line 27-29, there is a grammatical error in this sentence. Precisely, the use of the word “observing” deserves to be examined again.

2. Line 44, the more accurate expression is immunomodulatory or immune-modulatory. Moreover, there is a comma missing after [4].

3. Line 61-62, “a main … undergoes sulfation” as a parenthesis is inappropriate. Please rewrite this sentence.

4. In section 2.4, the authors should briefly supplement methods based on their experiments.

5. In section 3.5, the authors used the serum for biochemical analysis. But in Line 148, they used the plasma. Please confirm the usage carefully.

6. Line 399-402, the authors discussed that sulfation may enhance the bioactivity of polysaccharides. Is there a correlation between the sulfated sites and the bioactivity of polysaccharides? Please put some explanation about this.

Reviewer 3 Report

The manuscript of the article is very good, but some small corrections should be made.

1) The numbers in Figures 2, 3 and 5  should be larger as they are hard to see at the moment.

2) The quality of Figure 6 is too poor.
